# Mechanism of Inosine Monophosphate Degradation by Specific Spoilage Organism from Grass Carp in Fish Juice System

**DOI:** 10.3390/foods11172672

**Published:** 2022-09-02

**Authors:** Dapeng Li, Shuai Zhuang, Yankun Peng, Yuqing Tan, Hui Hong, Yongkang Luo

**Affiliations:** 1College of Food Science and Technology, Shanghai Ocean University, Shanghai 201306, China; 2College of Food Science and Nutritional Engineering, China Agricultural University, Beijing 100083, China; 3College of Engineering, China Agricultural University, Beijing 100083, China

**Keywords:** aquatic products, IMP degradation, specific spoilage bacteria, prokaryotic transcriptomic analysis, fish juice system

## Abstract

Microbial growth strongly affects the quality and flavor of fish and fish products. This study aimed to explore the role and function of grass carp-borne microorganisms in the degradation of inosine monophosphate (IMP) related compounds in a fish juice system during chill storage (4 °C. Prokaryotic transcriptomic analysis was used to explore the microbial contribution to metabolic pathways and related enzymes. The degree of microbial contribution was verified by the activity of enzymes and metabolite content. Collectively, there were multiple IMP relative product degradation pathways. *A. rivipollensis* degraded IMP by producing 5′-nucleotidase (5′-NT) while *S. putrefaciens* degraded IMP mainly by alkaline phosphatase (ALP). Hypoxanthine (Hx) was degraded to uric acid (Ua) induced by *P. putida* and *S. putrefaciens* mainly with producing xanthine oxidase (XOD), while *A. rivipollensis* almost could not produce XOD. This work can used as a guide and provide basic knowledge for the quality and flavor control of aquatic products.

## 1. Introduction

Nowadays, fish are becoming more and more important in the daily diets of human beings due to their ability to provide high-quality protein [1]. Fish provide more than 50% of animal protein in countries such as Bangladesh, Cambodia, Ghana, Indonesia, and some small island developing states [2]. However, fish is extremely perishable, as it is rich in protein and high in enzymatic activity. As a result, fish is usually stored at low temperatures to delay the enzymatic activity and microorganisms growing. A series of studies were conducted on the quality changes of fish during storage [3,4]. It was clear that the structure of fish bacterial groups changes dynamically with the changes in the environment of fish. There are a variety of microorganisms at the beginning of storage, while only a few microorganisms are dominant at the end of storage such as *Aeromonas* spp., *Photobacterium* spp., *Pseudomonas* spp. and *Shewanella* spp. [4,5]. These microorganisms participate in the spoilage process of fish and produce spoilage products, which are called specific spoilage organisms (SSOs) under specific storage conditions [6]. The SSOs vary in different fish species and under different storage conditions [6]. Silbande et al. [3] found that the specific bacteria for spoilage of tropical red drum stored at 4 °C were *Carnobacterium* spp. and *Brochothrix* spp. *Photobacterium phosphoreum* was the SSO for air-packed Atlantic cod at 4 °C [7]. The SSOs of freshwater fish were different from those of marine fish. *Pseudomonas* sp., *Aeromonas* sp. and *Shewanella* sp. are SSOs of freshwater fish such as silver carp, bighead carp, and grass carp [4,5,6,8].

Microbial growth can cause deterioration in fish quality, especially in flavor. An important reason for the popularity of fish among consumers is its delicious taste. Umami was the key point of the delicious taste in fish products. Numerous studies had shown that delicious amino acids, nucleotides, and umami peptides can provide the umami flavor [9,10,11,12]. Among them, inosine monophosphate (IMP) was the main flavor-presenting nucleotide in fish products [13]. The main source of IMP in fish was the degradation of adenosine triphosphate (ATP), which is an integral part of the postmortem changes in fish. ATP is degraded to adenosine diphosphate (ADP), adenosine monophosphate (AMP), and IMP by several enzymes, including ATPase, creatine kinase, and AMP deaminase. The degradation of ATP to IMP is usually rapid, ranging from a few hours to 2 days [14,15]. Several studies had shown that this process was mainly carried out by endogenous enzymes in fish and was less related to microorganisms [16,17,18,19].

The degradation of IMP and its associated products was closely related to microorganisms, mainly by various enzymes produced from microorganisms. The degradation of IMP and relative products during storage is the result of the combined action of multiple enzymes, and multiple enzymes may be involved in the same step of the reaction. Acid phosphatase (ACP), alkaline phosphate hydrolase (ALP), and 5′-nucleotidase (5′-NT) are the main enzymes that catalyze the degradation of IMP. Among them, 5′-NT is mainly present in the cell wall of bacteria and specifically catalyzes the degradation of IMP to hypoxanthine ribonucleoside (HxR) [13], while ACP and ALP are not specific enzymes but can also catalyze IMP to produce HxR. However, the key enzymes catalyzing IMP degradation are not clear. Vilas et al. [18] showed that as the number of bacteria increased, the rate of IMP degradation and hypoxanthine (Hx) formation increased greatly during storage. This may be related to the production of enzymes that catalyze IMP degradation and Hx formation by bacteria [16,17]. Some of the IMP may be degraded directly to Hx without the intermediate process of degradation to HxR, which may be due to the secretion of other enzymes by bacteria. A previous study showed that the degradation of IMP during the storage of carp may be related to the production of higher activity of ACP by microorganisms [17]. Purine nucleoside phosphorylase and inosine nucleosidase were the main enzymes controlling the conversion of HxR to Hx and were widely distributed in bacteria such as *Pseudomonas* sp. and *Shewanella* sp. Surette et al. [20] suggested that the production of inosine nuclease in fish may be due to the action of *Pseudomonas* sp. *Shewanella putrefaciens* and *Pseudomonas helmanticensis* had a strong ability to degrade HxR, which may be related to the production of inosine nucleosidase with higher activity by these bacteria [16]. In addition, the degradation of Hx to uric acid (Ua) is associated with xanthine oxidase (XOD). However, it is not clear which microorganisms produce XOD. In general, there is a lack of systematic studies on the effect of SSOs on the degradation of IMP and relative products during fish storage, and the abilities and pathways of each SSO are not clear. The source and ability of related enzymes in the degradation of IMP and relative products need to be further investigated.

In this context, we inoculated SSOs, including *Aeromonas rivipollensis, Pseudomonas putida*, and *Shewanella putrefaciens* from grass carp into sterile cooked grass carp juice to investigate: (1) the pathway of IMP degradation; (2) the relationship between IMP degradation and related enzyme activities; and (3) the mechanism of IMP degradation affected by SSOs. The implementation of this research was expected to provide a theoretical basis for the regulation of IMP in aquatic products. It is also an important guiding in the processing, storage, and transportation of aquatic products.

## 2. Materials and Methods

### 2.1. Strain Preparation

Three SSOs, including *A. rivipollensis, P. putida*, and *S. putrefaciens*, were isolated from spoiled grass carp flesh during 4 °C storage. They were preserved in the laboratory and were selected for activation in tryptic soy broth (TSB). Then 0.5 mL of the bacterial solution was inoculated in conical flasks containing 25 mL of TSB and incubated in a constant temperature water bath for 12 h (30 °C, 110 rpm). The total viable bacteria count was measured each hour. According to the results, the concentration of bacteria reached about 8.0–9.0 log CFU/mL after 9 h. The bacteria were incubated for 9 h as the final inoculum.

### 2.2. Fish Juice Preparation and Bacterial Inoculation

The sterile cooked fish juice was prepared by referring to the method of Dalgaard [21] with appropriate modifications. A total of 30 fresh grass carp (*Ctenopharyngodon idella*) were purchased from Beijing farmers’ markets with an average weight of 1560 ± 100 g and an average length of 40.3 ± 3.0 cm. Grass carp were transported alive to the laboratory in oxygen-filled polyethylene bags (15 °C) by the staff of the market transportation department. Grass carp were stunned by tapping them on the head. All practices performed in fish stunning were in strict accordance with the instruction of the World Organisation for Animal Health. Then they were scaled, gutted, cleaned with water, and filleted. After that, the red flesh and skin were removed, and the back flesh was taken. The fish back flesh was minced and weighed. The minced fish was mixed with deionized water in a ratio of 1:1, stirred and boiled for 10 min. The solids were separated from the liquid using a sterile gauge. The filtrate was then filtered through filter paper to remove microscopic particles. Finally, the filtrates were collected and dispensed into 60 mL per conical flask. They were sterilized at 121 °C for 20 min and the resulting sterile fish juice was stored in a refrigerator at 4 °C.

A total of 600 μL of bacterial inoculum (8.0–9.0 log CFU/mL) was added to 60 mL of fish juice and stored in a refrigerator at 4 °C for 10 d. A total of 10 mL of fish juice was taken under the sterile environment for subsequent studies every 2 days.

### 2.3. Determination of the Microbiological Number

The plate count method was used to determine the microbial count during the refrigeration of fish juice, following the method of Li, et al. [22] with some modifications. A total of 0.5 mL of sample was serially diluted with 4.5 mL of sterile 0.9% NaCl solution in a 10-fold gradient. Three appropriate dilutions were selected. A total of 100 μL of each of them was applied uniformly on plate count agar (PCA) plates. The plates were incubated in an incubator at 30 °C for 48 h and then counted. The microbiological number was expressed as log CFU/mL, and the minimum limit value was 2 log CFU/mL.

### 2.4. IMP-Related Compounds

The IMP-related compounds, including ATP, ADP, AMP, IMP, HxR, and Hx, were extracted according to Li, et al. [17] A total of 1 mL of fish juice sample was mixed with 2 mL of cold 10% perchloric acid (PCA) and leave to set for 5 min at 4 °C. The mixture was centrifuged at 1500× *g* for 5 min, and we collected the supernatant. A total of 2 mL of cold 5% PCA was used to wash the sediment, and it was centrifuged at 1500× *g* for 5 min. All the supernatant was collected in a 50 mL centrifuge tube. The pH of the supernatant was adjusted to 6.4–6.5 with 1 mol/L KOH. Then, the mixture was centrifuged at 1500× *g* for 5 min. The supernatant was collected and volumed to 10 mL.

The supernatant was filtered through a 0.22 μm membrane and the IMP-related compounds were analyzed by high-performance liquid chromatography (HPLC) (Shimadzu, LC-10 ATseries, Japan) equipped with an SPD-10A (V) detector. The separation was run on a COSMOSIL 5C18-PAQ column (4.6 mm × 250 mm) (Nacalai Tesque, Inc., Kyoto, Japan) with phosphate buffer (0.05 mol/L, pH 6.9) pumped at 1 mL/min at a temperature of 25 °C. The detection was monitored at 254 nm and compared to standards (Sigma-Aldrich Trading Co., Ltd., Shanghai, China).

The content of uric acid (Ua) in the fish juice was determined using a Ua assay kit (Nanjing Jiancheng Bioengineering Institute, Nanjing, China). A sample of 1 mL fish juice was centrifuged at 550× *g* for 5 min. The supernatant was collected and used to determine the Ua content. The Ua content was determined following the method of assay kits.

### 2.5. The Activities of ACP, ALP, 5′-NT, and XOD

ACP, ALP, and 5′-NT activity were determined using the ACP, ALP, and 5′-NT assay kits (Nanjing Jiancheng Bioengineering Institute, Nanjing, China) respectively, following the method described by Li, et al. [17]. XOD activity was determined using an XOD assay kit (Solarbio Life Sciences Institute, Beijing, China). A sample of 1 mL of fish juice was centrifuged at 550× *g* for 5 min. The supernatant was collected and used to determine the enzyme activities. The enzyme activities were determined following the method of assay kits and expressed as units per liter (U/L).

### 2.6. Prokaryotic Transcriptomic Analysis

#### 2.6.1. RNA Extraction, Library Construction, and Sequencing

A total of 10 mL of fish juice was centrifuged at 550× *g* for 5 min, and the sediment was collected. They were stored at −80 °C for the prokaryotic transcriptomic analysis. Total RNA was extracted from the sediment using TRIzol^®^ reagents according to instructions (Invitrogen). The genomic DNA was removed using DNase I (TaKara). After that, RNA quality was determined using an Agilent 2100 Bioanalyzer and the quantification was quantified using ND-2000 (NanoDrop Technologies). High-quality RNA samples (OD260/280 = 1.8~2.0, OD260/230 ≥ 2.0, RIN ≥ 6.5, 23S:16S ≥ 1.0, concentration ≥ 100 ng/μL, total ≥ 2 μg) were used for subsequent library construction.

The TruSeqTM RNA sample preparation Kit from Illumina (San Diego, CA, USA) was used to perform the RNA-seq transcriptome library. The rRNA was removed using the Ribo-Zero Magnetic kit (epicenter). The mRNA was broken into small fragments of about 200 bp randomly. Double-stranded cDNA was synthesized by reverse transcription using random primers (Illumina) and SuperScript double-stranded cDNA synthesis kit (Invitrogen, CA, USA) using mRNA as the template. In the synthesis of the second strand of cDNA, dUTP was used instead of dTTP for synthesis. The synthesized double-stranded cDNA was patched into flat ends by adding End Repair Mix. PCR amplification was then performed with Phusion DNA polymerase (NEB) for 15 cycles. After quantification with TBS380 (Picogreen), RNA-seq double-end sequencing was performed using Illumina HiSeq X Ten (2 × 150 bp).

#### 2.6.2. Bioinformatics Analysis

Bioinformatics analyses were performed using data generated from the Illumina platform. All analyses were performed using the Majorbio Cloud Platform (Available online: www.majorbio.com (accessed on 25 May 2021)) from Shanghai Majorbio Bio-pharm Technology Co., Ltd.

Bowtie2 (Available online: http://bowtie-bio.sourceforge.net/bowtie2/index.shtml (accessed on 25 May 2021)) was used to map the reference genome. The Blast method and Rfam database (Available online: http://rfam.xfam.org (accessed on 25 May 2021)) were used to assess rRNA contamination rates. The gene and isoform abundances from single-end or paired-end RNA-Seq data were quantified by RSEM (Available online: http://deweylab.github.io/RSEM (accessed on 25 May 2021)). TPM, the number of reads from a particular transcript per million reads, was used to measure the expression levels of genes. The Gene Ontology (Available online: http://www.geneontology.org (accessed on 25 May 2021)) project and Kyoto Encyclopedia of Genes and Genomes (KEGG, Available online: http://www.genome.jp/kegg (accessed on 25 May 2021)) were used to analyze the classification and biological functions of genes.

### 2.7. Statistical Analysis

All experiments were carried out in triplicate. All data were subjected to one-way analysis of variance (ANOVA) followed by the Duncan method with a significance level of 5%.

## 3. Results and Discussion

### 3.1. Number of Microorganisms

As shown in Figure 1, the total viable counts (TVC) in the control group were below the minimum limit value (2 log CFU/mL) throughout the storage process, indicating that the samples were prepared and stored without contamination. During storage, *A. rivipollensis*, *P. putida*, and *S. putrefaciens* grew to 8.9 log CFU/mL, 9.5 log CFU/mL and 9.2 log CFU/mL on 10 d, respectively. Moreover, there was no significant difference between the three species on 8 d. Therefore, the samples were taken on 8 d and used for prokaryotic transcriptomic analysis.

### 3.2. The Prokaryotic Transcriptomic Analysis

Figure 2 shows the results obtained from the preliminary analysis of prokaryotic transcriptomic among the three bacteria. The pathway of IMP-related compounds degradation in the cooked fish juice system induced by *A. rivipollensis* (green), *P. putida* (red), and *S. putrefaciens* (blue) was set out in Figure 2A. IMP in fish mainly comes from the degradation of ATP and a variety of enzymes, including nucleoside-diphosphate kinase [EC:2.7.4.6], pyruvate kinase [EC:2.7.1.40], and adenylate kinase [EC:2.7.4.3], which were involved in this complex biochemical process. Different microorganisms were involved in different biochemical pathways and the production of related enzymes. During fish storage, the degradation of ATP to AMP takes a short time, usually occurring within 1 day. In this study, we found that a variety of enzymes were involved in this process, which was a reversible reaction. Transcripts of these related enzyme genes were also identified in three SSOs. However, in the actual fish storage process, due to the small number of microorganisms at this stage (usually <3 log CFU/g), these enzymes mainly come from the fish itself. The research of Liu et al. [16] and Li et al. [17] also showed that this stage had little relationship with microorganisms. After that, AMP was further degraded into IMP. Alasalvar et al. [23] suggested that this effect was completely autolytic in sea bass (*Dicentrarchus labrax*). The degradation of AMP to IMP was mainly due to the action of AMP deaminase, and this biochemical reaction was irreversible. However, none of these three bacteria had the transcription and expression of this enzyme gene. This was consistent with our previous research showing that AMP deaminase was mainly derived from fish flesh itself rather than produced by microorganisms [17].

IMP was an important freshness substance for fish, and its presence and maintenance contribute to the flavor of fish products [24,25]. During the fish storage, the degradation of IMP to Hx was slower than the degradation of ATP to IMP [26]. This may be caused by the fact that the degradation of IMP requires the growth of microorganisms and the accumulation of related enzymes. IMP was mainly degraded to HxR by the action of enzymes, which was an irreversible reaction. Numerous studies have shown that phosphate hydrolases were the main enzymes involved in this reaction [13,15,16,17]. Among them, ACP [EC: 3.1.3.2], ALP [EC: 3.1.3.1], and 5′-NT [EC: 3.1.3.5] were considered the main enzymes involved in the degradation of IMP, all of which were phosphate hydrolases [13]. In the present study, the transcription of the 5′-NT gene was found mainly in *A. rivipollensis* and *P. putida*, and the transcription level of *A. rivipollensis* was higher than that of *P. putida* (*p* < 0.05). The transcription of the ALP gene was found mainly in *S. putrefaciens* and was much higher than that of the other two SSOs (*p* < 0.05). the transcription of the ACP gene was not found in these three SSOs. This result suggests that different SSOs were involved in IMP degradation by different enzymes; *A. rivipollensis* and *P. putida* were involved in IMP degradation in fish juice mainly by producing 5′-NT, while *S. putrefaciens* was involved mainly by producing ALP. In addition, a new enzyme, Inosine kinase [EC:2.7.1.73], mainly derived from *A. rivipollensis* and *S. putrefaciens*, was found to be involved in the conversion of IMP and HxR as well. This reaction was reversible. The function of this enzyme has been less reported previously. Its capacity and role need to be further investigated.

HxR and its degradation products (Hx, Xa, Ua) were considered spoilage substances, and their presence seriously brings down the quality and flavor of fish products. The degradation of HxR to Ua is considered to be closely related mainly to microbial activity [18,25,27]. During fish storage, HxR is mainly degraded by IMP, which has a slightly bitter taste, but was much lower than Hx. Hong et al. [13] reported that the conversion of HxR to Hx was caused by the purine-nucleoside phosphorylase enzyme. Liu et al. [16] found that Inosine nucleosidase was involved in the degradation of HxR during cold storage of bighead carp, and was mainly derived from *A. sobria*, *P. helmanticensis*, and *S. putrefaciens*. These views were echoed by the present study. Based on the prokaryotic transcriptomic analysis, two enzymes are involved in the HxR degradation. Purine-nucleoside phosphorylase [EC:2.4.2.1] was derived from three SSOs and dominated the reversible conversion reaction of HxR to Hx. On the other hand, purine nucleosidase [EC:3.2.2.1], which dominates the irreversible degradation reaction of HxR, was only derived from *P. putida* and *S. putrefaciens*. The transcript level of the purine nucleosidase gene was significantly higher in *S. putrefaciens* than that of *P. putida* (*p* < 0.05). However, it was not known which of the two enzymes plays the dominant role in the degradation of HxR and further studies were needed.

The degradation of Hx to Xa and Ua was mainly a function of xanthine dehydrogenase [EC:1.17.1.4]. From Figure 2, it can be seen that all of these three SSOs could produce xanthine dehydrogenase. However, the transcript and expression levels of xanthine dehydrogenase genes were significantly lower (*p* < 0.05) in *A. rivipollensis* than in *P. putida* and *S. putrefaciens*. It was suggested that the ability of *A. rivipollensis* to degrade Hx to Xa and Ua was much lower than that of the other two SSOs. In contrast, Liu, et al. [16] detected xanthine oxidase (i.e., xanthine dehydrogenase) activity in bighead carp inoculated with *A. sobria*. Such different results suggested that gene expression may differ significantly between different bacterial species of the genus *Aeromonas* spp.

The metabolism of IMP-related products during fish storage mainly follows the process of ATP → ADP → AMP → IMP → HxR → Hx → Xa → Ua [8,13]. A few investigators have indicated that other degradation pathways for IMP-related products may exist. Vilas, et al. [19] built up a model for the biochemical degradation of inosine monophosphate in hake (*Merluccius merluccius*) and suggested a possible existence of a pathway for direct conversion of IMP to Hx. In the present study, besides the most studied process of ATP → ADP → AMP → IMP → HxR → Hx → Xa → Ua, some other pathways for the degradation of IMP and relative products, including the pathway of direct conversion of IMP to Hx, were also found. In detail, AMP can be degraded to adenosine via 5′-NT. On the one hand, adenosine can be converted to HxR via the action of adenosine deaminase [EC:3.5.4.4]. On the other hand, adenosine can be degraded to adenine via the action of purine-nucleoside phosphorylase [EC:2.4.2.1] and purine nucleosidase [EC:3.2.2.1] and further degraded to hx via adenine deaminase [EC:3.5.4.2]. AMP can also be directly degraded to adenine by purine-5′-nucleotide nucleosidase [EC:3.2.2.-] and adenine phosphoribosyltransferase [EC:2.4.2.7]. briefly, three metabolic pathways, including ① AMP → adenosine → HxR → Hx, ② AMP → adenosine → adenine → Hx, and ③ AMP → adenine → Hx, may exist with the involvement of the three SSO. In addition, IMP can interconvert with Hx via hypoxanthine phosphoribosyltransferase [EC:2.4.2.8], corroborating the conjecture of Vilas, et al. [19]. However, the role and mechanisms of these metabolic processes in fish storage need to be further investigated.

### 3.3. The Activity of IMP-Related Enzymes

From the previous prokaryotic transcriptome analysis, it was found that multiple enzymes were involved in IMP degradation in cooked fish flesh juice inoculated with three SSO. However, whether their activity was related to the transcriptional level of the relevant enzyme genes needs to be investigated. Therefore, the present study continued to investigate the activities of three phosphate hydrolases (ACP, ALP, and 5′-NT) affecting IMP degradation and XOD for Hx degradation of cooked fish flesh juice during 4 °C storage.

The changes in the activities of the four enzymes are summarized in Figure 3. Since the fish juice had undergone a sterilization and passivation process at 121 °C for 20 min before inoculation with the three SSOs, we can assume that the change in enzyme activity in the fish juice was mainly due to the SSO inoculation. Therefore, we can assume that the changes in the enzyme activities in the fish juice were mainly caused by the inoculated SSOs. The enzyme activities of the control group shown in Figure 3 can be considered systematic errors. The activity of ACP and ALP differed between fish. ACP activity was higher in horse mackerel (*Trachurus japonicus*), while ALP activity was higher in gurnard (*Lepidotriga microptera*) [28]. Wang et al. [29] found that the activity of ACP was much higher than that of ALP in grass carp and catfish. In contrast, as grass carp flesh was used to make cooked fish flesh juice, the ACP that came from the fish itself could be excluded. The activity of ACP was significantly lower than that of ALP in this study, which may be related to the less transcription and expression of ACP genes in these three SSOs. The ALP activity in fish juice inoculated with *S. putrefaciens* increased substantially during the first 4 days of storage and then maintained fluctuating changes, which was associated with no substantial change in the number of *S. putrefaciens* after day 4 (Figure 1). This indicates that the changes in ALP activity were closely related to the changes in the number of *S. putrefaciens*. Similarly, the 5′-NT activity in inoculated *A. rivipollensis* fish juice was closely correlated with changes in the number of *A. rivipollensis*. In addition, the ALP activity in the juice of fish inoculated with *S. putrefaciens* was significantly higher than the other two SSOs throughout the storage process. This result corroborates the transcription and expression levels of the ALP gene in Section 3.2. The 5′-NT activity in the juice of fish inoculated with *A. rivipollensis* was significantly higher than that of the other two SSOs. Although the results of the prokaryotic transcriptome analysis indicated that *P. putida* also had higher transcription and expression levels of the 5′-NT gene, *P. putida* may lack the 5 ‘-NT activators, making the 5′-NT activity lower.

Similar to the pattern of activity change of ALP, the activity of XOD showed a pattern of increase followed by smooth fluctuation during the storage of fish juice inoculated with *S. putrefaciens*. It reached 9.9 U/L on the 6th day of storage. In contrast, the enzymatic activity of XOD was detected in *P. putida* only on day 6. Combined with the growth pattern of *P. putida* in Figure 1, it was speculated that XOD production and accumulation were not available until *P. putida* reached the stabilization stage. Based on the prokaryotic transcriptomic analysis results, there were higher transcription and expression levels of the XOD gene in *S. putrefaciens*, followed by that in *P. putida*. There was almost no transcription of the XOD gene in *A. rivipollensis*. Correspondingly, XOD activity was highest in *S. putrefaciens*, while it was 0 in *A. rivipollensis*. Liu et al. [16] reported that XOD activity could be detected in fish inoculated with *A. sobria*, *P. helmanticensis*, *S. putrefaciens*, and *Ac. bohemicus*. Among them, *A. sobria* and *Ac. bohemicus* had lower XOD activity. This suggested that it had different XOD gene expression among different *Aeromonas* spp. In addition, it was speculated that *Aeromonas* spp. has a lower ability to produce XOD compared to *Pseudomonas* sp. and *Shewanella* sp., implying a weaker ability to degrade Hx.

### 3.4. IMP-Related Compounds

The variation in the content of IMP-related compounds was more indicative of the ability of SSO to act on them. The content of ATP, ADP, and AMP in the fish juice was almost 0. This may be due to the thermal degradation of IMP-related substances induced by the high-temperature sterilization treatment of the fish juice [30]. The contents of IMP, HxR, Hx, and Ua in fish juice during storage were shown in Figure 4.

The initial IMP content in fish juice was 2.10 × 10^3^ μmol/L. With the increase in storage time, the IMP content in fish juice inoculated with *A. rivipollensis* decreased rapidly to 0.0 on day 6. The IMP content in fish juice inoculated with *S. putrefaciens* decreased more slowly to 1.41 × 10^3^ μmol/L on day 10 of storage. The IMP content in the juice of fish inoculated with *P. putida* remained almost unchanged throughout the storage period. Interestingly, this was in accordance with the previous changes in phosphate hydrolysis activity. This suggests that both ALP and 5′-NT can degrade IMP to HxR, which is similar to the results of the previous study. In addition, combining the activity changes of ALP and 5′-NT during storage, it was presumed that 5′-NT has a strong ability to degrade IMP.

The initial HxR content in the fish juice was 2.00 × 10^3^ μmol/L, which was close to the IMP content. The HxR content in the juice inoculated with the three SSO decreased continuously as the storage time increased. The HxR content in the juice of fish inoculated with *A. rivipollensis* and *S. putrefaciens* decreased to 0 on days 6 and 4, respectively, while the HxR content of fish juice inoculated with *P. putida* decreased slowly, reaching 0.94 × 10^3^ μmol/L on day 10 of storage. *Aeromonas salmonicida* in the Atlantic salmon (*Salmo salar* L.) had a strong ability to degrade HxR [7], whereas *Pseudomonas* sp. and *S. putrefaciens* were the main contributors to the accumulation of HxR and Hx in freshwater fish flesh [8,16]. The different findings may be due to the differences among bacterial species.

The levels of Hx and Ua showed interesting changes during storage. In fish juice inoculated with *A. rivipollensis* Hx reached 5.53 × 10^3^ μmol/L on day 6 and changed smoothly thereafter. This corresponds to a decrease in HxR to 0 on day 6. Interestingly, the content of Ua, on the other hand, did not show any increase during storage. This suggested that *A. rivipollensis* did not have the ability to degrade Hx to Ua which correlated with its almost non-existent production of XOD (Section 3.2, Figure 2 and Figure 3). On the contrary, the Hx content in the juice of fish inoculated with *S. putrefaciens* increased slowly in the pre-storage period, reaching 2.54 × 10^3^ μmol/L on the fourth day of storage and gradually decreasing thereafter. In contrast, the Ua content increased in the juice inoculated with *S. putrefaciens*. Combined with the results of prokaryotic transcriptome analysis and XOD enzyme activity, it indicated that *S. putrefaciens* produced a large amount of XOD and contributed to the degradation of Hx to Ua. The Hx content in the juice of fish inoculated with *P. putida* slowly increased from day 4, corresponding to the time of the decline of HxR. Although *P. putida* also produced XOD, the low content of its substrate Hx led to a slow rise in Ua. Liu et al. [16] found that *P. helmanticensis* and *S. putrefaciens* had a stronger ability to degrade Hx and produce Ua than *A. sobria*. Combined with the present study, we speculate that the weaker ability of *A. rivipollensis* to degrade Hx was mainly due to its lower production of XOD, the degrading enzyme of Hx. On the contrary, *P. putida* and *S. putrefaciens* have a stronger ability to degrade Hx and accumulate Ua. Due to the different SSO species and different storage conditions for marine fish, different enzymes and metabolic pathways for IMP degradation may be found. More research on the degradation of IMP-related substances in marine fish is needed in the future.

## 4. Conclusions

The study aimed to determine the effect and mechanism of action of three SSOs (*A. rivipollensis*, *P. putida,* and *S. putrefaciens*) on the degradation of IMP and its relative products. This study showed that the three SSOs differ in their mechanisms of action on IMP-related product degradation. *A. rivipollensis* degrades IMP through the production of 5′-NT while *S. putrefaciens* degrades IMP mainly through ALP. *P. putida* and *S. putrefaciens* degraded Hx to the final product Ua by producing XOD. In contrast, *A. rivipollensis* produced fewer XOD, which resulted in its weak ability to accumulate Ua. Furthermore, this study also showed that SSOs play a role in a variety of other biochemical reaction pathways in addition to the currently studied main biochemical process of ATP → ADP → AMP → IMP → HxR → Hx → Xa → Ua. The mechanism and roles of different pathways in the IMP degradation induced by SSO need to be further investigated. The understanding of the mechanisms of different SSO degradation IMP can be a guide for quality control and flavor regulation of aquatic products, which can be utilized by the food industry.

## Figures and Tables

**Figure 1 foods-11-02672-f001:**
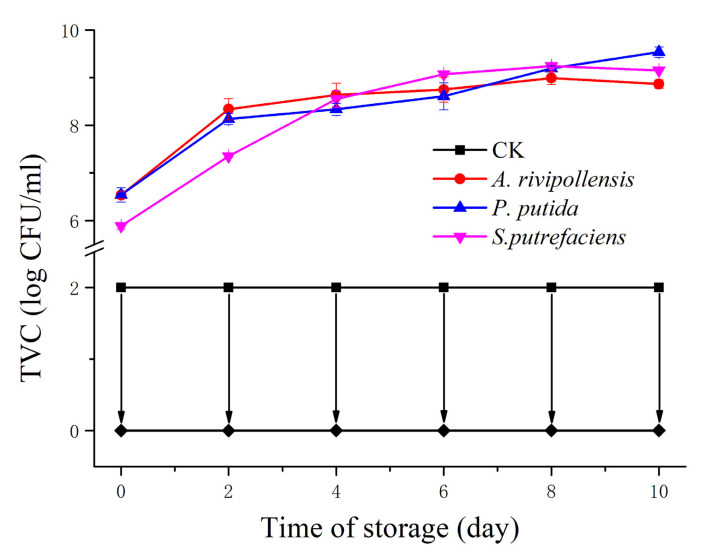
Changes in the total viable counts (TVC) of grass carp juice inoculated with different spoilage bacteria during 4 °C storage. CK: Not inoculated.

**Figure 2 foods-11-02672-f002:**
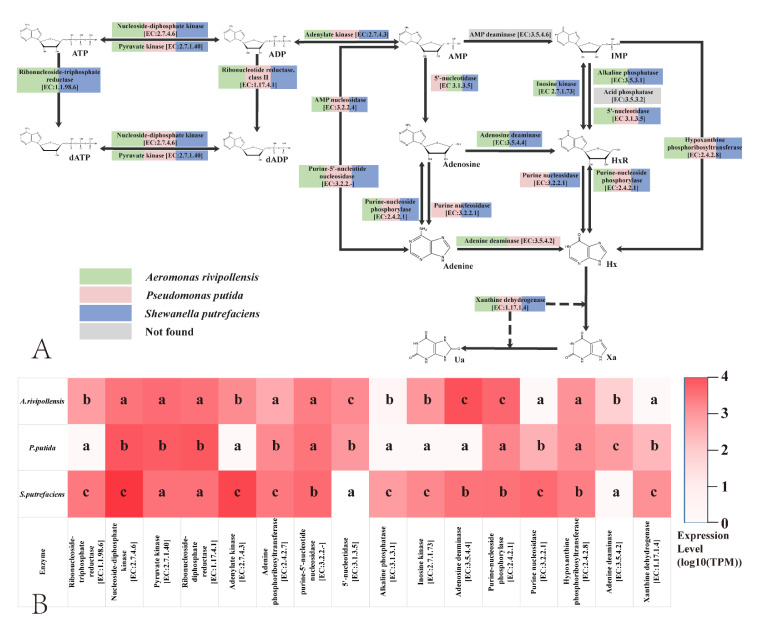
Microbial contributions in degrading purine nucleotide compounds. (**A**) Purine nucleotide compounds pathways in grass carp juice induced by specific spoilage bacteria against the GO and KEGG database. (**B**) Gene expression analysis of specific spoilage bacteria-induced in grass carp juice revealed by gene transcriptomics results. The same lowercase in a column indicates no significant differences (*p* > 0.05).

**Figure 3 foods-11-02672-f003:**
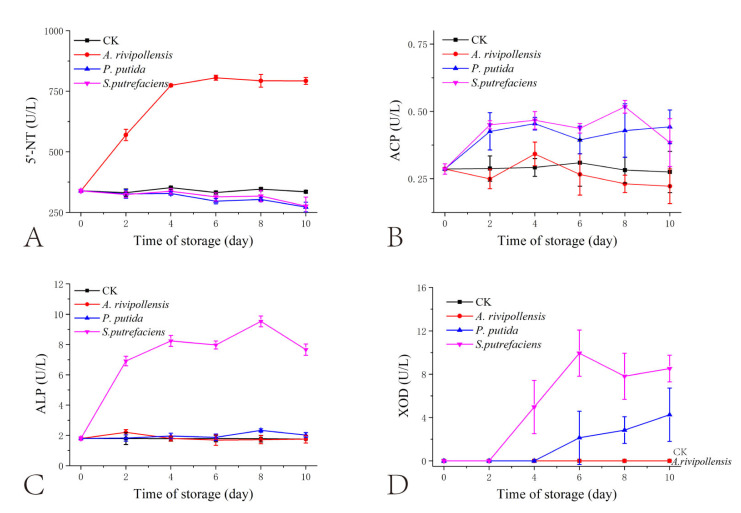
Changes in related enzyme activities of grass carp juice inoculated with different spoilage bacteria during 4 °C storage. (**A**) 5′-nucleotidase (5′-NT); (**B**) acid phosphatase (ACP); (**C**) alkaline phosphatase (ALP); (**D**) xanthine oxidase (XOD). CK: Not inoculated.

**Figure 4 foods-11-02672-f004:**
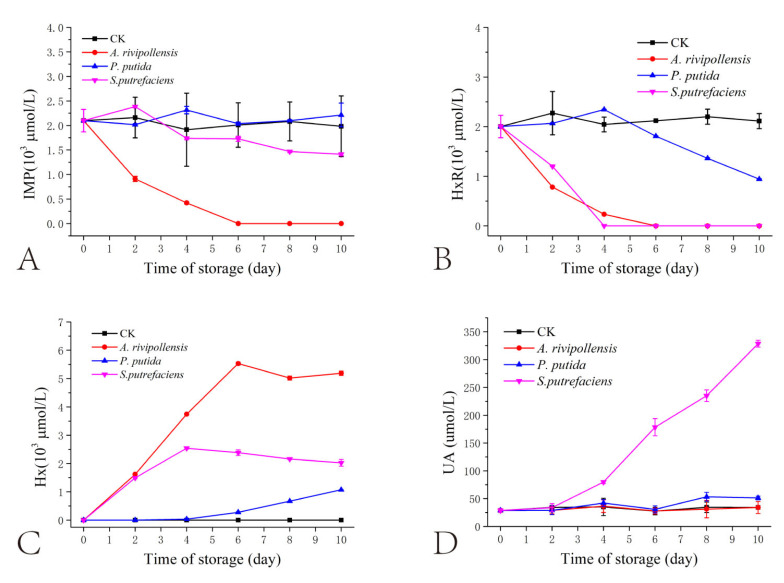
Changes in IMP-related compounds contents of grass carp juice inoculated with different spoilage bacteria during 4 °C storage. (**A**) Inosine monophosphate (IMP); (**B**) Hypoxanthine ribonucleoside (HxR); (**C**) Hypoxanthine (Hx); (**D**) Uric acid (Ua).CK: Not inoculated.

## Data Availability

Data is contained within the article.

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
