# Peer review of "Mechanism of Inosine Monophosphate Degradation by Specific Spoilage Organism from Grass Carp in Fish Juice System"

_foods, 2022, doi:10.3390/foods11172672_

Round 1

Reviewer 1 Report

Dear Editor,

This manuscript aims to identify the inosine monophosphate degradation in fish by certain spoilage microorganisms found in fish. It is a well-planned experiment that unfortunately suffers from mediocre use of the English language. Other than that, this is an interesting article.

Specific points that need the author’s attention are highlighted in the attached file, along with the relevant questions or proposed edits

Author Response

Dear reviewer:

Thank you for the comments. We have revised the manuscript and provided a point-by-point response in the attachment. Each of the responses was written following your notes. The revised manuscript was uploaded to the system. 

Sincerely, 
Dapeng Li

Reviewer 2 Report

The manuscript includes an interesting study, well written and presented.

Abstract

Line 13: Provide temperature.

Lines 20-21: Perform ( … new ideas … ?).

Introduction

Line 39: Perform reference format. This mistake is done in several parts of the manuscript, too.

Material and methods

Line 102: Provide Latin name of the species.

Lines 192-206: Eliminate.

Figures

Figure 1: Perform units in Y axis (mL of what ?). Explain CK abbreviation.

Figure 3: Perform units in Y axis (L of what ?). X axis: Time of what ?

Figure 4: Perform units in Y axis (L of what ?). X axis: Time of what ?

Discussion/Conclusions

Carp was chosen as fish model. Could the results be different if a marine species had been chosen ? Please, include some discussion on this in the Discussion and Conclusions sections.

It is well known that some fish species (i.e., flat ones) do not undergo transformation from inosine to hypoxanthine. This also ought to be taken into account when doing generalisations.

Reviewer 3 Report

Review titled “Mechanism of inosine monophosphate degradation by specific spoilage organism from grass carp in fish juice system”

In general: The work deals with the degradation ATP via IMP in fish juice caused by spoilage bacteria. Here, the activity of known spoilage enzymes by three different bacterial species is studied by transcriptomic analysis. A major point of criticism is that only one strain per bacterial species was tested in this work, aspects of strain diversity should be discussed! It is noted that during storage very high levels of germs develop in the fish juice, at which fish is normally no longer fit for consumption. Have you carried out a sensory evaluation (e.g. odor, color) of the samples? This is necessary for an evaluation of the determined bacterial counts and the measured enzyme contents!

It is mandatory to revise the text of the results and discussion section. In the results section, only your own results may be discussed to explain the figures. Discussion means: the own results are compared to those of the literature! Avoid duplication of statements in revision. Alternatively, you can combine the items results and discussion.

1. Abstract

Line 20: What kinds of ideas you mean. Please make a concrete statement!

2. Introduction

Line 30: There are any legal rules acc. storage of fresh fish at the Chinese retail level? In Europe fish have to be store on ice (storage at melting temperature).

Line 38: Do you mean fish species? Please clarify!

Line 60: What is the idea of this purpose?

Line 64: What means HxR, Hx, XOD? Please describe these abbreviations once in the text!

Please describe exactly why you are performing this study. What is the novelty compared to other publications. What is the hypothesis that underlies your studies?

3. Material and Methods

Line 93: Which is the source of these bacteria (fish, meat, …?)?

Line 141: What means Ua??

Line 164: The (the)

4. Results

Line 209: There cannot be a bacterial count of 0 cfu/g for diluted samples. Please specify a limit value below the detection limit.

5. Discussion

Is missing, but mandatory! You could discuss it together with the results (see general comments)

6. Conclusion

Line 426 – 429: Are there any concrete correlations between flavor and the degradation of IMP by bacteria (please discuss)? You have not investigated this yourself, have you?

Round 2

Reviewer 3 Report

I`m agrre with the changes im the manuscript.